# Population Space–Time Patterns Analysis and Anthropic Pressure Assessment of the Insubric Lakes Using User-Generated Geodata

**Alberto Vavassori †** , **Daniele Oxoli \*,†** and **Maria Antonia Brovelli**

Department of Civil and Environmental Engineering, Politecnico di Milano, Piazza Leonardo da Vinci 32, 20133 Milano, Italy; alberto.vavassori@polimi.it (A.V.); maria.brovelli@polimi.it (M.A.B.)
\* Correspondence: daniele.oxoli@polimi.it
† These authors contributed equally to this work.

**Abstract:** Human activities are one of the main causes of lake-water pollution and eutrophication. The study of human pressure around lakes is of importance to understand its effects on the lakes natural resources. Social media data is a valuable space–time-resolved information source to detect human dynamics. In this study, user-generated geodata, namely users' location records provided by the Facebook Data for Good program, are used to assess population patterns and infer the magnitude of anthropic pressure in the areas surrounding the Insubric lakes (Maggiore, Como and Lugano) between Northern Italy and Southern Switzerland. Patterns were investigated across different lakes' neighbouring areas as well as seasons, days of the week, and day hours in the study period May 2020–August 2021. Two indicators were conceived, computed and mapped to assess the space–time distribution of users around lakes and infer the anthropic pressure. The highest pressure was found around lakes Maggiore and Como coastal areas during weekends in summer (up to +14% average users presence than weekdays in winter), suggesting tourism is the primary accountable reason for the pressure. Contrarily, around lake Lugano, the population dynamic is mostly affected by commuters or weekly workers, where the maximum pressure occurs during weekdays in all seasons (+6.6% average users presence than weekends). Results provide valuable input to further analyses connected, for example, to the correlation between human activities and lake-water quality and/or prediction models for anthropic pressure and tourism fluxes on lakes that are foreseen for the future development of this work.

**Keywords:** lakes; anthropic pressure; population dynamics; social media; user-generated geodata

## 1. Introduction

Natural lakes play a fundamental role for the environment and communities that developed on their shores. In fact, lakes act as reservoirs that supply fresh water for agricultural and domestic usage. They contribute to mitigating the effects of floods and droughts by storing large volumes of water. Lakes often support local economic growth as they allow for manifold recreational and tourism activities that take advantage of landscape attractiveness and water resources. Lakes contribute to the preservation of biodiversity by hosting ecosystems and valuable habitats for different plant and animal species. Finally, they positively affect local climate conditions by buffering extreme temperatures in their neighbouring areas [1].

Increasing urbanisation, industrial production, farming and livestock production and tourism activities along lake shores often result in widespread environmental problems due to an increasing load of discharged pollutants that is directly responsible for lakes' water-quality degradation [2]. This is especially true where water treatment facilities are not adequate for adsorbing these sharp or occasional increases in pollutants loads [3]. By focusing on tourism, recreational activities such as swimming, boating, angling, etc.,

directly impact lake water. For instance, boats can damage aquatic plants and stir up sediments that may promote algal growth. Actually, any tourist-related activities occurring within the lake catchments have an effect on the natural resources. For example, the construction of holiday villages, sports centres and other tourism facilities permanently affect the lake ecosystem by introducing land-use changes [4] that reduce the capacity of natural waterways to intercept nutrients and often lead to an increased pollutants load [3,5]. Impact assessments of tourism and, in general, human activities on lake ecosystems have been tackled in the literature. Sun and Liu [6] studied the effect of tourism on the West Lake Basin in China, observing a positive correlation between pollution indexes and variables connected to tourism activities, such as tourists count, tourism economic income and tourist garbage. Markogianni et al. [7] analysed the effect of human activities, land use, and soil loss on the water quality of Plastira Lake in Greece, pinpointing the tourism industry as one of the most impactful activities. Evidence of the positive correlation between human activities and water pollution is also found for the Insubric lakes considered in this study (see Figure 1) [8–10]. To summarise, population growth and human activities are clearly identified as the main cause of inner-water eutrophication and pollution worldwide [3].

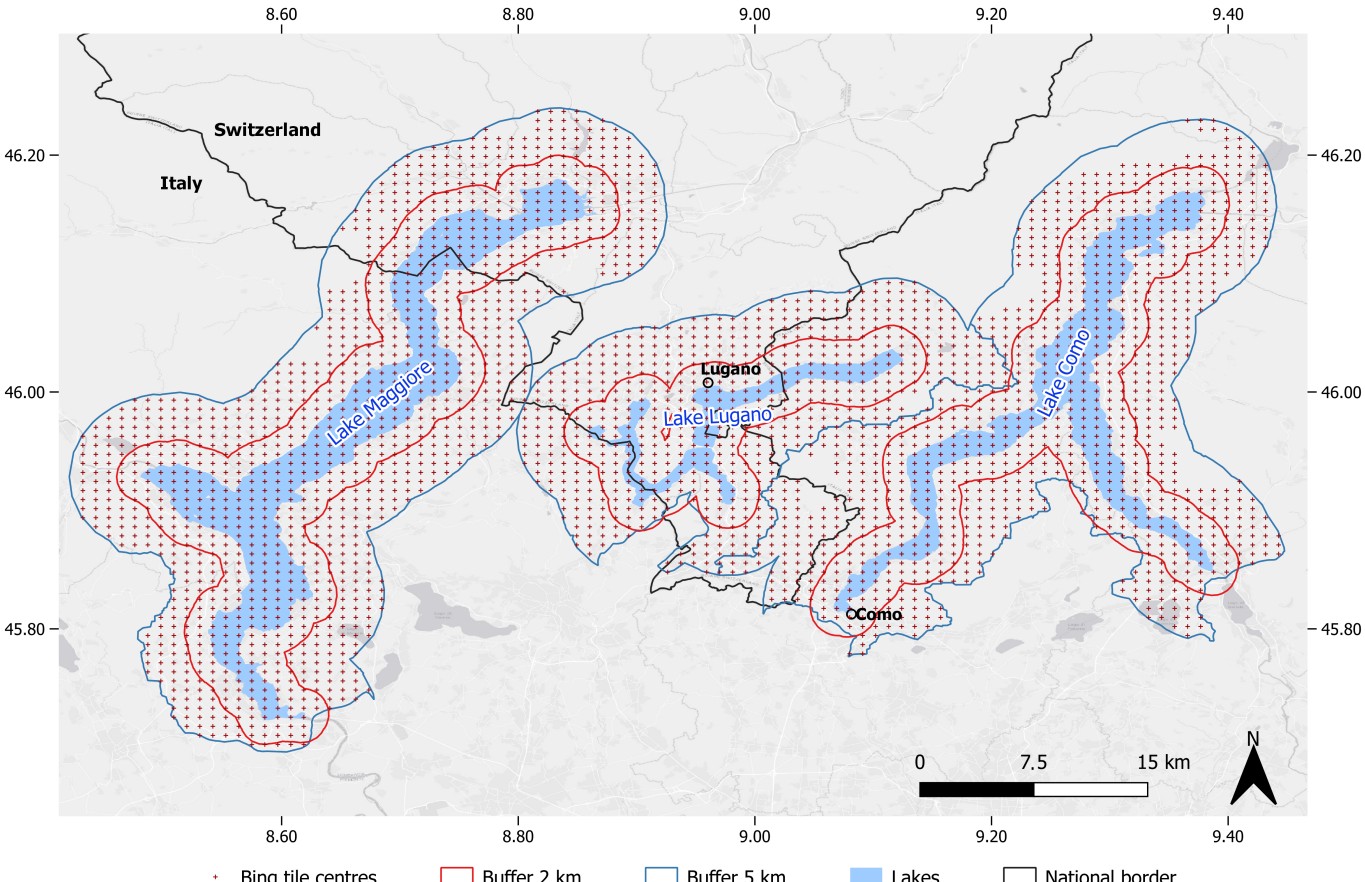

**Figure 1.** Study areas (red and blue outlined polygons), Insubric lakes (from left to right: Lake Maggiore, Lake Lugano and Lake Como) and Bing tile grid centres (red dots) for the considered Facebook Population Maps collection. Como and Lugano are the most populated urban centres of the study region. CRS: WGS84. Basemap data: Esri Gray (light) © Esri.

In view of the above, the study of human presence and mobility around lakes deserves thorough investigation. Nevertheless, disaggregated and space–time resolved data, which are demanded by local population dynamics analyses, are often lacking or limited by the geographical extent, amount and quality of information [11]. In the recent past, social media, user-generated and crowdsourced georeferenced data have emerged in scientific applications on tourism and human mobility [12]. Thanks to their abundance and granularity

they are often able to complement traditional data sources such as governmental statistic reports, surveys and monitoring campaigns [13]. Social media data have been widely employed to assess human mobility in both urban and natural environments. Relevant applications focused on the use of geolocated posts or users' location records retrieved from social media, such as Twitter, Foursquare and Facebook, to analyse mobility and human mobility patterns. This type of information has been used, for example, to assess the space–time distribution of human movements and tourism flow between different location types [14,15], and characterise patterns in urban mobility trajectories [16]. Jiang et al. [17] analysed the distribution of tweets to understand how population density varies between the city centres and the peripheral areas. Ebrahimpour et al. [18] used Weibo data to cluster users' positions in urban areas. Social media data have also been employed for assessing tourist mobility in natural areas [13] and to detect local and foreign tourists' preferred destinations [19].

With this in mind, the purpose of the present study is to investigate population space–time patterns and trends and derive indicators of anthropic pressure on a lake's district for detecting locations that suffer from occasional or systematic excesses of human presence with respect to their resident population. The study area is the region of Insubric lakes between Northern Italy (lakes Maggiore and Como) and Southern Switzerland (Lake Lugano). This study is carried out within the framework of the SIMILE project (Informative System for the Integrated Monitoring of Insubric Lakes and their Ecosystems), funded by the Interreg program of the European Union with the aims of strengthening the coordinated management and monitoring sub-alpine lakes between Italy and Switzerland [20]. The work was carried by leveraging user-generated geodata provided by the Facebook Data for Good initiative [21]. The data used in this study consist of users' location records, from which users' population patterns across space and time are analysed to uncover and describe significant local differences across the study area in different seasons as well as weekdays and day hours.

This emerging data source has been recently employed, for example, to analyse changes in human mobility during COVID-19 lockdown in different countries [22,23] and population displacements after large-scale disasters [24], including fires [25]. In a previous work published by the co-authors of this paper Oxoli and Brovelli [26], which represents a preliminary case study for the presented methodology, these data were used to infer visitor fluxes in natural parks of the Insubria region.

The outcomes of the statistical and exploratory analyses are summarised in graphs, tables and maps allowed to identify and quantify users' population space–time patterns which result as markedly different between the Italian and Switzerland lakeshore areas. The ultimate goal of this study is to infer the population dynamic around the study lake areas to empower the understanding of its influence on the lake natural ecosystems. The obtained results are promising, supporting further investigations of the correlation between population space–time patterns and lake-water quality, which are envisaged for the future development of this work.

The remainder of this paper is structured as follows. Section 2 describes the datasets from the Facebook Data for Good initiative that have been primarily adopted in this study. Section 3 illustrates the methods employed for data processing and analysis, whereas in Section 4 results are presented and discussed. Finally, in Section 5, conclusions and future directions of this work are reported.

## 2. Data and Case Study Definition

### 2.1. User-Generated Geodata from Facebook

The user-generated geodata considered in this study are retrieved from the Facebook Data for Good initiative [21]. The main purpose of this initiative is to provide policymakers, researchers and humanitarian organisations with insightful geospatial information which may be used to prepare for, respond to, and recover from crisis events (such as disasters or a disease outbreak) or, more generally, to empower studies of population behaviours and

displacement. Among the available Facebook for Good initiative products, the Facebook Disaster Maps provides records of Facebook users' locations in time. Spatial aggregation of individual records is performed by the data provider [27] in order to ensure users' privacy protection [28]. Among the Facebook Disaster Maps datasets, the one called Facebook Population Maps was considered in this study. This dataset provides counts of users with location services enabled on their mobile devices who stay within a specific location in a defined time interval [29]. Data locations refer to pixels of geographic grids derived from the Bing tile system [30] and the count of users per pixel is provided with a temporal resolution of 8 h, namely 12 a.m.–8 a.m., 8 a.m.–4 p.m., and 4 p.m.–12 a.m. Greenwich Mean Time (GMT). This space–time data aggregation prevents any individual user tracking operation while allowing only for summary local patterns analysis at different locations of the study area. Bing tiles with users' count values lower than 10 are removed from the dataset by the provider and, therefore, are not considered in the presented analyses. The dataset is distributed in CSV format in the WGS84 geographic reference system and can be downloaded by registered users from the Facebook Data for Good partner portal [31].

### 2.2. Study Areas and Period

The collection of Facebook Population Maps for the Insubria region was activated in mid-2020 upon a request of the authors to the data provider. The considered study period was 6 May 2020–31 August 2021. Facebook users' population counts were provided with a spatial resolution of (approximately) 850 m, which corresponds to the Bing tile level 7 [30], and a temporal resolution of 8 h. The available spatial resolution of the data grid depends mainly on the extent of the study region for which a collection is activated. This means that the larger the study region, the lower is the resolution of the datasets. This trade-off is due to the computational time required by the Facebook servers to generate the dataset within the defined time slot of 8 h.

The study areas were selected as follows. Two different buffers around lakes Maggiore, Como and Lugano were extracted as reported in Figure 1. The first buffer of 2 km width was defined to focus on lakeshores, which are likely to be more influenced by tourism. Due to the particular morphology of the considered lakes, which are characterised by narrow shores bounded by steep mountain slopes, such small buffers allowed us to delimit only the areas most affected by lake tourism. The second buffer of 5 km width includes large residential districts that are located beyond the lakeshore and in mountain areas and which are expected to be marginally—or even not at all—affected by lake tourism flows. The 5 km buffer was then adjusted to exclude sub-areas external to the lake catchments so that only the users population directly affecting the lake's water resources was considered in the analysis. The sizes of the different study areas are reported in Table 1.

**Table 1.** Extensions [km$^2$] of the considered study areas.

| Lake | 2 km Buffer | 5 km Buffer |
| :---: | :---: | :---: |
| Maggiore | 501.8 | 1033.9 |
| Como | 420.8 | 864.9 |
| Lugano | 214.7 | 476.8 |

### 2.3. Facebook Data Representativeness

A critical concern when dealing with user-generated geodata, such as the Facebook Population Maps, is data representativeness. In this work, this is intended as the ratio between the average Facebook users' counts computed in the study areas and the actual resident population.

The average users' population was defined as the mean of the Facebook users' counts during night-time (12 a.m.–8 a.m. GMT) in the low tourism seasons (fall and winter) from 25 October 2020 to 27 March 2021. The selection of this specific time interval, for the years 2020 and 2021, is connected to the Italian and Swiss daylight saving time which results

in 1 h offset in the fall and winter seasons from their summertime zone (GMT + 1). The underlying assumption is that as lakes tourism is at its most concentrated during spring and summer (high tourism season), the night-time population during the low season is best for representing the resident Facebook users population [32].

The actual resident population was instead derived from the WorldPop raster dataset [33] that provides estimation for the resident population at 100 m resolution for each country in the World. The motivation behind the use of this particular dataset instead of traditional census data is twofold. The main reason is that the boundaries of the study region do not overlap with any existing census tracks, which is the reference area unit adopted by the national statistic bureaus to distribute census statistics. The second reason is that the Insubria region is located across the international border between Italy and Switzerland, affecting the availability of consistent census data covering the whole area.

The mean Facebook users count was computed both at each lake area and the whole study region and then divided by the correspondent actual resident population derived from the WorldPop dataset. Results from the assessment of Facebook data representativeness are reported in Table 2. The average representativeness for the whole region resulted as close to 4%, with small differences among the three lakes and the two buffer areas. It is worth noticing that Facebook data are not expected to represent the whole population in a geographic area as they are generated only by Facebook users with location services enabled on their mobile devices [27], which in turn represents only a portion of all Facebook users.

**Table 2.** Facebook data representativeness for the study areas, expressed as the ratio [%] between the mean night-time Facebook users' count during the low tourism seasons and actual resident population retrieved from the WorldPop dataset [33].

| Lake | Buffer 2 km | Buffer 5 km |
|---|---|---|
| Maggiore | 4.32% | 4.19% |
| Como | 3.83% | 4.04% |
| Lugano | 4.57% | 4.52% |
| whole region | 4.21% | 4.23% |

## 3. Data Processing and Analysis

The analysis was carried out by aggregating data both in space and time. Space aggregation separately considered the two buffer areas for the three lakes (Maggiore, Como and Lugano). Afterwards, time aggregation was performed to distinguish between high and low tourism seasons, days of the week (weekdays/weekends), and day hours divided into time slots of 8 h according to the original temporal resolution of the data. The temporal aggregation considered low tourism season as the period between 25 October 2020 and 27 March 2021 and the high season as the rest of the year, according to that which is explained in Section 2. Finally, weekends included Saturday and Sunday whereas weekdays were the remaining days.

Data analysis focused on the computation of summary statistics and exploration of differences between Facebook users' population counts for the different aggregation time periods over each identified study area. A preliminary comparison of results from the two buffer areas was also performed to investigate the assumption of the difference in population patterns between lakeshores and surrounding areas due to tourism activities.

The assessment of anthropic pressure on lakes was accomplished by designing and computing a synthetic local indicator. The indicator was expressed as the ratio between the 90th percentile computed from the complete time series of Facebook users' population and a baseline value for each location in the dataset. The baseline values at each location were computed as the mean of the Facebook users' population counts in low tourism season night-time, which is the same strategy adopted for the data representativeness assessment (see Section 2). The 90th percentile was arbitrarily chosen as the peak value to account not

only for maximum values of the time series, which might be affected by local events in time such as festivals, sporting events, etc., but to portray more frequent peaks of users' presence with the aim of better approximating the persistent pressure affecting each location.

An additional consideration on the time of occurrences of the local peaks pressure was carried out. In this case, maximum values of Facebook users' population at each location and their month and day-of-the-week of occurrences were also mapped to explore their space–time distribution. Both the outlined anthropic pressure indicator and the magnitudes of the maximum values with their times of occurrence were finally mapped by providing a synoptic insight into the space–time of the analysed phenomenon within the study region.

Data processing and analysis were performed using popular Python data analysis libraries including Dask, Pandas, GeoPandas, and Matplotlib [34]. The data downloaded from the Facebook Data for Good partner portal consisted of 1446 CSV files (i.e., three files per day of the study period) with a 13 GB memory. The files were read in a single Pandas Dataframe and custom functions were developed for data aggregation, summary statistics and indicators computation, and results plotting. Results of the analysis are discussed and summarised using graphs and maps in the following section.

## 4. Results and Discussion

### 4.1. Facebook Users' Population Patterns Exploration

A preliminary comparison between daily mean Facebook users' population counts for the two buffer areas around each lake (see Figure 2a,b) was carried out. The purpose was to point out characteristic patterns for lakeshores (2 km buffer), which are expected to be more influenced by tourism, and surrounding areas (5 km buffer) that also include large urban centres. The magnitude of mean Facebook users counts follows the size of the different study areas (see Table 1). However, the experiment was intended to describe global trends and patterns rather than infer actual population values, which are known to be biased by the lower representativeness of the Facebook data as reported in Table 2.

In Figure 2b, it can be observed that the lakes Maggiore and Como show similar patterns for the 2 km buffer area with peaks registered during the high tourism season. Differences are mostly located around mid-summer periods for the 5 km buffer area, especially for the Lake Como (see Figure 2a). An explanation of the above may be the following. On the one hand, both lakes are relevant tourist destinations suffering from an increased human presence concentrated in the high season. On the other hand, Lake Como's 5 km buffer area includes most of the Como City residential district (see Figure 1), thus showing a marked reduction in human presence during typical Italian and Southern Switzerland holiday periods such as mid-summer and, secondarily, the Christmas period. The Lake Maggiore 5 km buffer area does not include large urban centres, thus showing a comparable pattern with its 2 km buffer area.

A significantly different situation is observed for Lake Lugano in the Swiss side of the study region. The daily mean Facebook users' population counts for both buffer areas are less influenced by the tourism season and, unlike the other two lakes, small increases are detected only in the middle of spring and fall seasons. A marked reduction in human presence is registered during holiday periods (i.e., mid-summer and Christmas) in both buffer areas (see Figure 2a,b). The latter is aligned with the one of Lake Como's 5 km buffer area and, indeed, this buffer area for Lake Lugano includes Lugano City residential district thus showing a population dynamic not mostly affected by tourism. The Lake Lugano 2 km buffer area follows the same pattern of its 5 km buffer areas, therefore suggesting Lake Lugano shores are a less popular destination for recreational and tourism activities compared to lakes Maggiore and Como.

According to the above insights, further explorations of space–time patterns focused on the 2 km buffer areas so that the anthropic pressure mostly imputable to tourism was investigated.

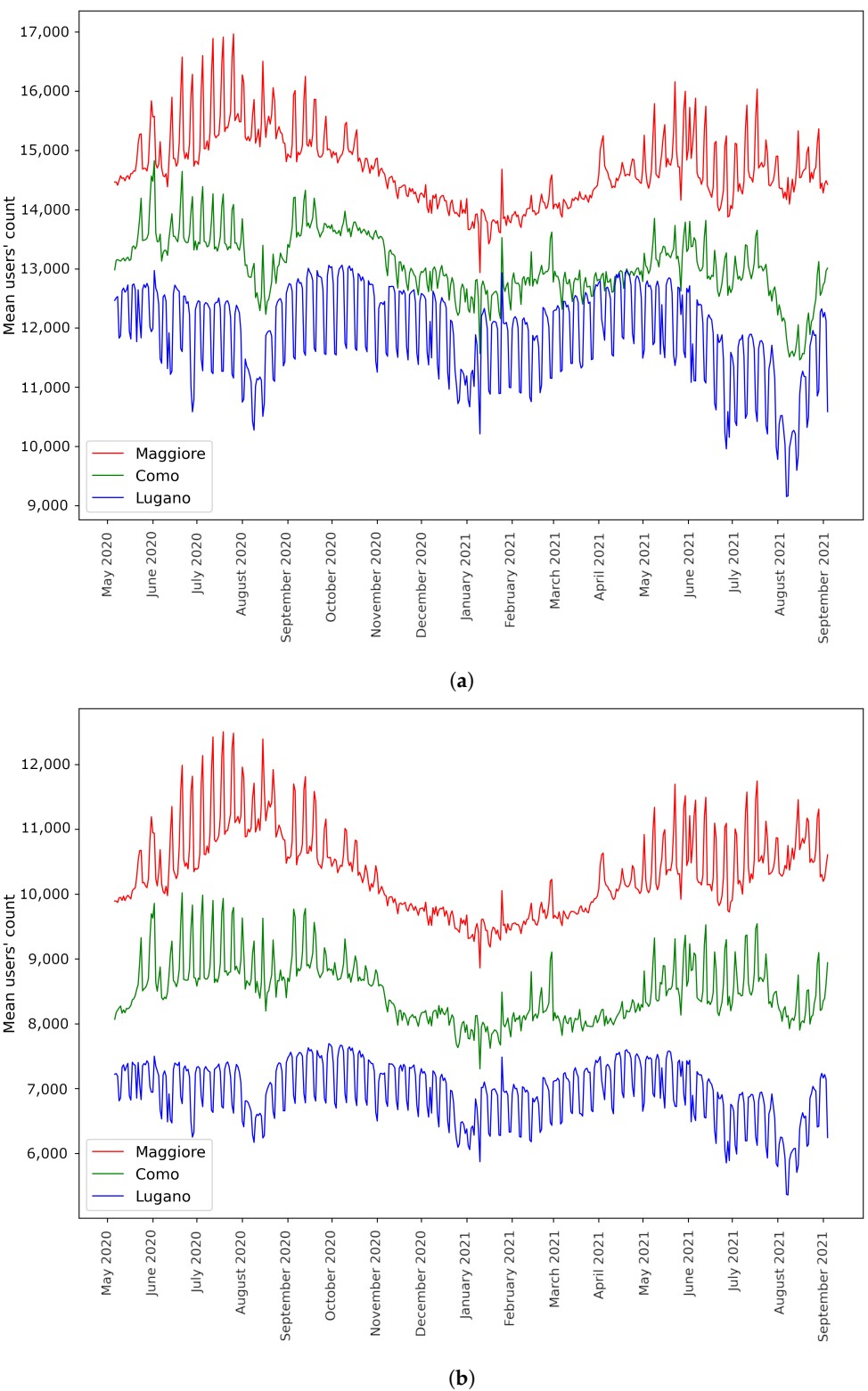

**Figure 2.** Time series of the daily mean Facebook users' population counts by lake within the 5 km (**a**) and 2 km (**b**) buffer areas surrounding the considered lakes for the study period 6 May 2020–31 August 2021.

Furthering the data exploration, the monthly means of the Facebook users' population counts were computed for the 2 km buffer areas of each lake. The monthly means were extracted by distinguishing between weekdays and weekends to allow for description of the influences of short-stay tourism (i.e., day trips or weekends) which is expected to

happen mostly during non-working days. Results are reported in Figure 3. The monthly distribution of Facebook users' population counts is also displayed through boxplots [35] in Figure 4.

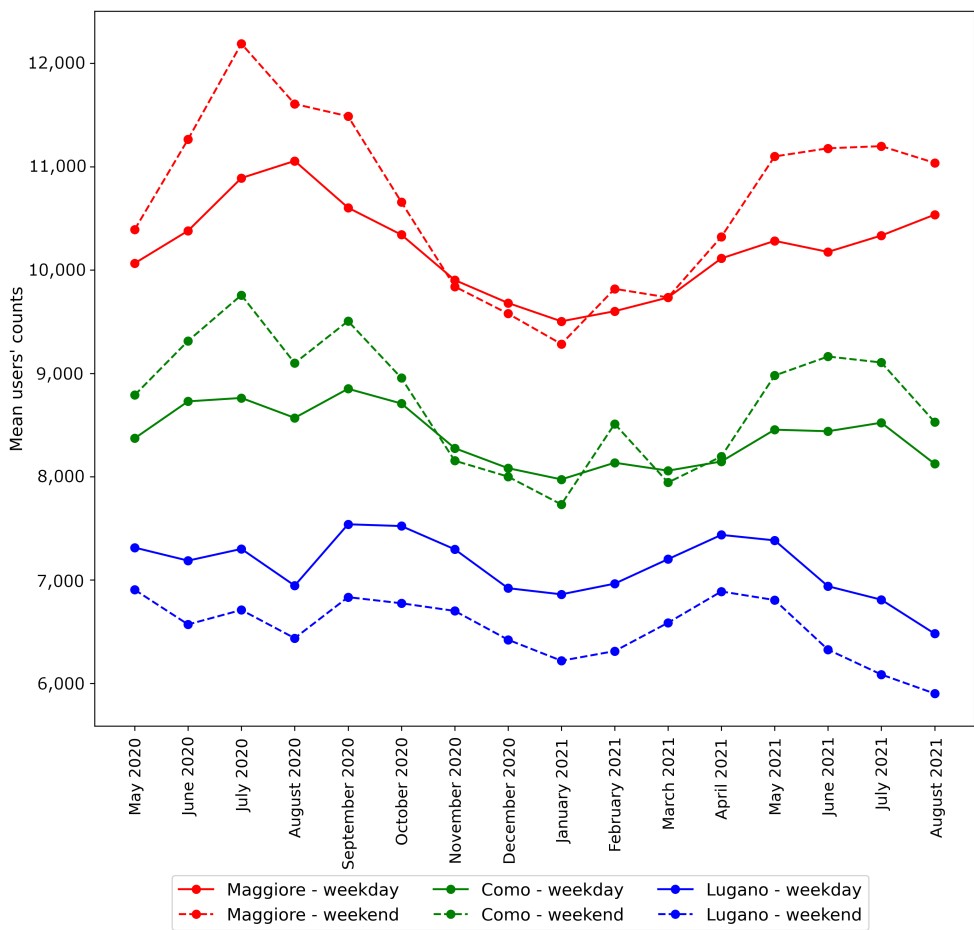

**Figure 3.** Time series of the monthly mean Facebook users' count by weekends (dashed line) and weekdays (solid line) for the three lakes.

The trends exhibited by the monthly means (see Figure 3) provide additional pieces of evidence of the similarity of the patterns between Maggiore and Como lakes while remarking their differences with Lake Lugano. Accordingly, peaks of user presence are observed during high tourism season, with a significant enhancement during weekends, only within Maggiore and Como lake areas. These patterns enforce the consideration of the latter lakes as popular tourist destinations, especially for short stays during non-working days. In contrast, Lake Lugano showed an opposite pattern where monthly means during weekends are constantly lower than weekdays, with peaks mostly placed outside the central periods of the high tourism season. These differences may reflect the fact that the Lake Lugano area is known to be affected by significant fluxes of frontier workers [36,37] so that the contribution of tourism is hidden due to the relatively lower amount of users reaching its shores for leisure and recreational activities.

By looking at the monthly distribution of Facebook users' population counts by weekends and weekdays in the boxplots of Figure 4, the above considerations are additionally confirmed. The deviation from the medians of Facebook users' population counts resulted as limited for Maggiore and Como lakes during weekdays (see Figure 4a) whereas higher dispersion was registered during weekends in the high tourism season (see Figure 4b). This behaviour may be reasonably justified by the presence of visitors concentrated during weekends in the high tourism season, which is expected to be less constant than the

presence of resident users' population. Concerning Lake Lugano, the distributions of users' counts concentrated around their median values with higher dispersion observed during weekdays in holiday periods (i.e., mid-summer and Christmas, see Figure 4a). This behaviour suggests the users' population pattern is driven by the presence of occasional visitors rather than frontier workers and resident users' population as opposed to the other months.

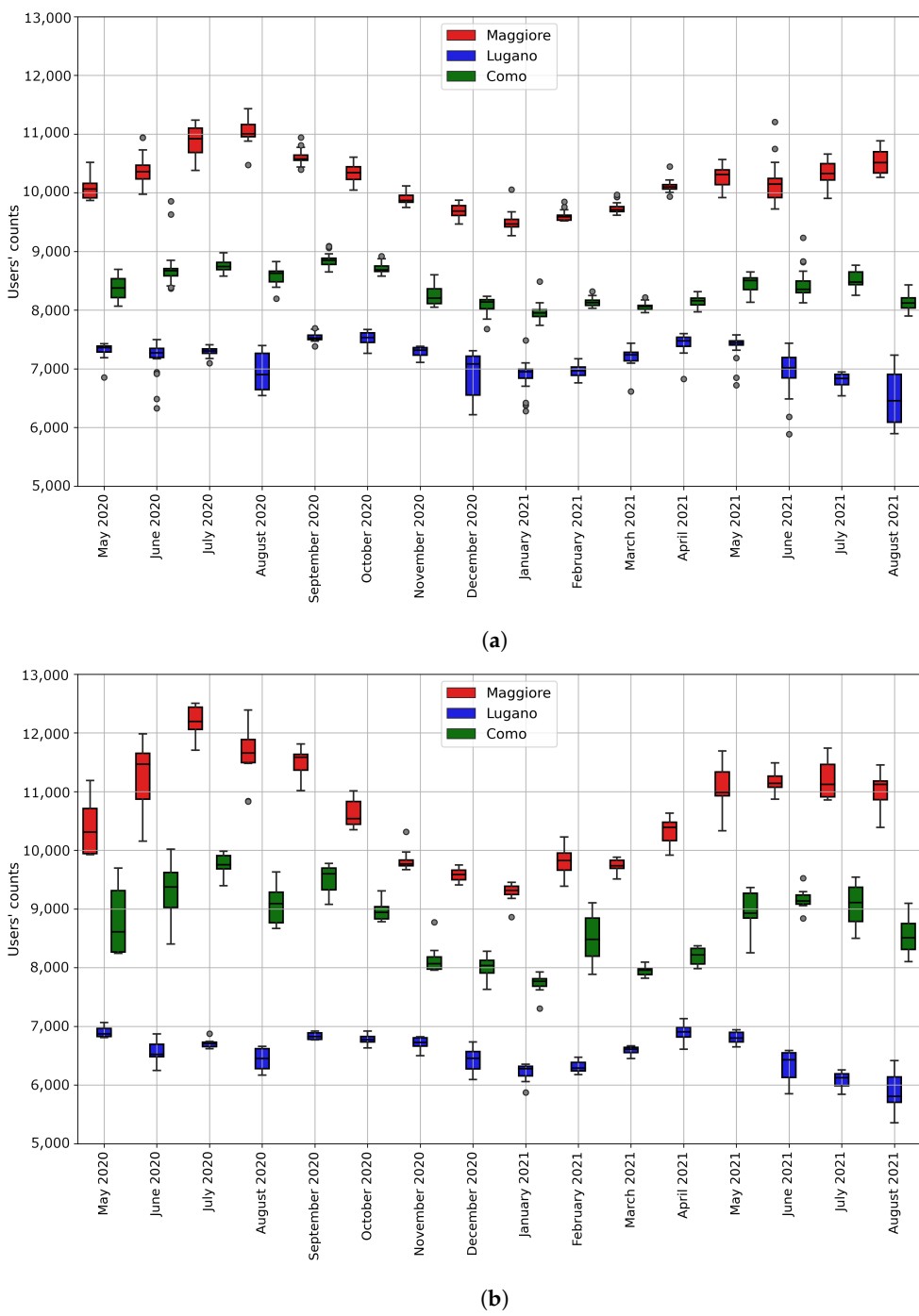

(**a**)

(**b**)

**Figure 4.** Boxplots of the Facebook users' count distributions per month by weekdays (**a**) and weekends (**b**) for the three lakes. Each box is delimited by the first (Q1) and third (Q3) quartiles and the horizontal line inside the box is the median. The two lines extending from the box (whiskers) describe the variability outside Q1 and Q3. Outliers are represented by points beyond the whiskers.

In all cases, only a few outliers appeared from the boxplots. A simple test on the presence of significant outliers was performed through the comparison of monthly means and medians of Facebook users' population counts for the three lake areas. No meaningful differences were observed between the two statistics as reported in Table 3, suggesting results from the monthly mean patterns exploration were not biased by outliers and extreme values.

**Table 3.** Monthly mean and median values of the Facebook users' counts by lake areas. Relative differences [%] between the two statistics are normalised by using the medians as a reference.

| | Mean | | | Median | | | (Mean–Median)/Median | | |
|---|---|---|---|---|---|---|---|---|---|
| | **Maggiore** | **Como** | **Lugano** | **Maggiore** | **Como** | **Lugano** | **Maggiore** | **Como** | **Lugano** |
| May 2020 | 10,166 | 8500 | 7188 | 10,065 | 8376 | 7300 | 1.00% | 1.48% | −1.53% |
| June 2020 | 10,616 | 8885 | 7023 | 10,392 | 8686 | 7207 | 2.15% | 2.29% | −2.55% |
| July 2020 | 11,226 | 9019 | 7149 | 11,091 | 8799 | 7291 | 1.21% | 2.50% | −1.95% |
| August 2020 | 11,233 | 8740 | 6782 | 11,064 | 8663 | 6659 | 1.53% | 0.89% | 1.85% |
| September 2020 | 10,839 | 9025 | 7351 | 10,629 | 8859 | 7509 | 1.98% | 1.87% | −2.10% |
| October 2020 | 10,433 | 8780 | 7305 | 10,427 | 8751 | 7462 | 0.06% | 0.33% | −2.10% |
| November 2020 | 9884 | 8240 | 7119 | 9852 | 8175 | 7237 | 0.32% | 0.78% | −1.63% |
| December 2020 | 9654 | 8061 | 6792 | 9662 | 8109 | 6786 | −0.07% | −0.59% | 0.09% |
| January 2021 | 9433 | 7896 | 6654 | 9438 | 7905 | 6841 | −0.06% | −0.12% | −2.73% |
| February 2021 | 9663 | 8243 | 6778 | 9602 | 8156 | 6904 | 0.63% | 1.06% | −1.82% |
| March 2021 | 9734 | 8029 | 7043 | 9726 | 8042 | 7159 | 0.08% | −0.16% | −1.62% |
| April 2021 | 10,168 | 8160 | 7291 | 10,115 | 8173 | 7425 | 0.52% | −0.17% | −1.81% |
| May 2021 | 10,546 | 8624 | 7197 | 10,379 | 8541 | 7404 | 1.61% | 0.97% | −2.80% |
| June 2021 | 10,442 | 8632 | 6776 | 10,239 | 8449 | 6872 | 1.99% | 2.17% | −1.39% |
| July 2021 | 10,585 | 8691 | 6599 | 10,479 | 8631 | 6734 | 1.01% | 0.70% | −2.01% |
| August 2021 | 10,681 | 8242 | 6313 | 10,680 | 8180 | 6243 | 0.01% | 0.76% | 1.11% |

Finally, the mean Facebook users' counts by day of the week and 8 h time slots during both high and low tourism seasons were computed as shown in Figure 5. All lake areas show a constant mean users' population during night time (12 a.m.–8 a.m. GMT) on weekdays while increases are registered during day-time (8 a.m.–4 p.m. and 4 p.m.–12 a.m. GMT). The patterns for the different lakes follow the ones observed in the previous comparisons. The users' population showed peaks during weekends for Maggiore and Como lakes (which are more pronounced during the high tourism season) while Lake Lugano had the opposite trend. Weekday peaks for the Lake Lugano area were clustered into the 8 a.m.–4 p.m. hours slot that overlaps with working hours. This enforces the assumption that the Lake Lugano weekdays population dynamic is mostly driven by frontier workers, as explained before. Weekday peaks for Maggiore and Como lakes were instead distributed between day-time hours slots whereas weekends peaks were clustered in the 8 a.m.–4 p.m. hours slot. This suggests daily visitors have a significant effect on the users' population, especially on Sundays during the high tourism season.

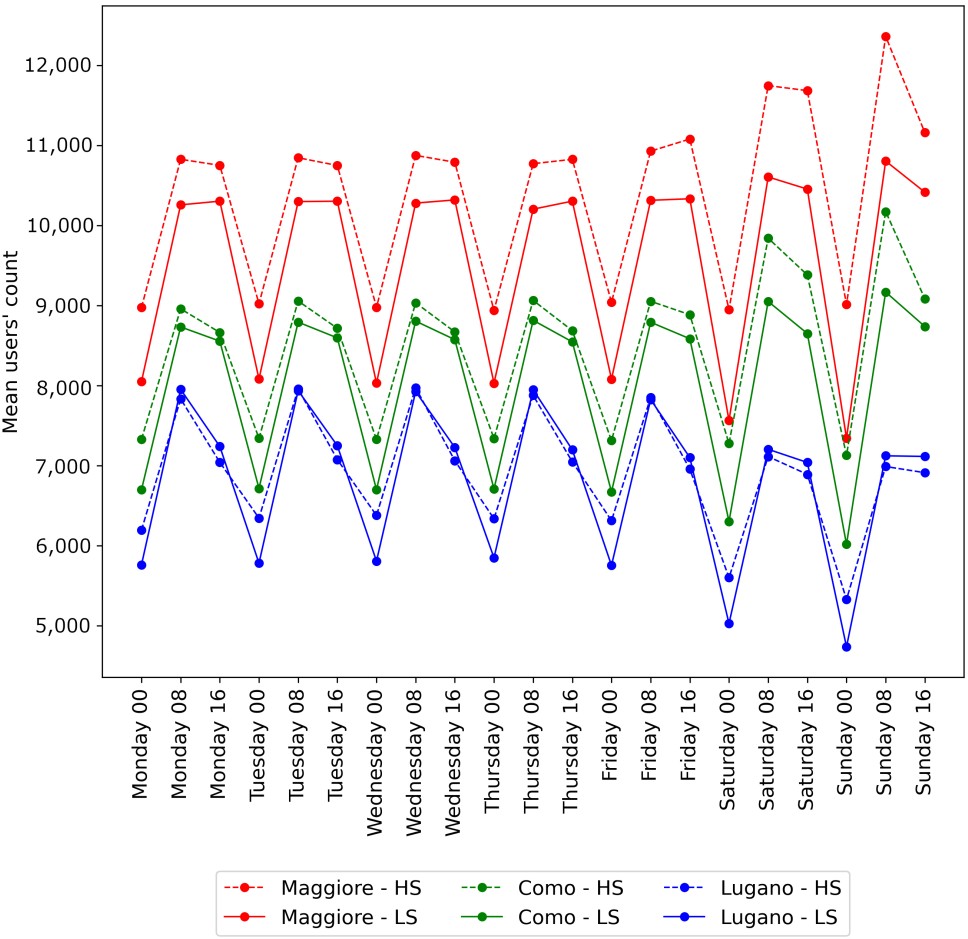

**Figure 5.** Variation of mean Facebook users' count per day and hour, during high (HS) and low (LS) tourism season.

### 4.2. Anthropic Pressure Assessment

Anthropic pressure was assessed both in space and time by mapping the indicators described in Section 3. Results are included in Figures 6 and 7.

Figure 6 includes the spatial distribution of values of the ratio between the 90th percentile computed from the complete time series of Facebook users' counts and the baseline population at each location (Bing tile centres). The resulting map allowed for the identification of the highest pressure exerted on each considered location within lake areas and provided a measure of the occasional users' presence increases—most likely related to tourism—with respect to the baseline population. The highest values of the indicator occurred on locations belonging to the central Lake Como areas, and both Southern-west and Northern Lake Maggiore. Not surprisingly, these are the areas including some of the most famous tourism hotspots of the study region.

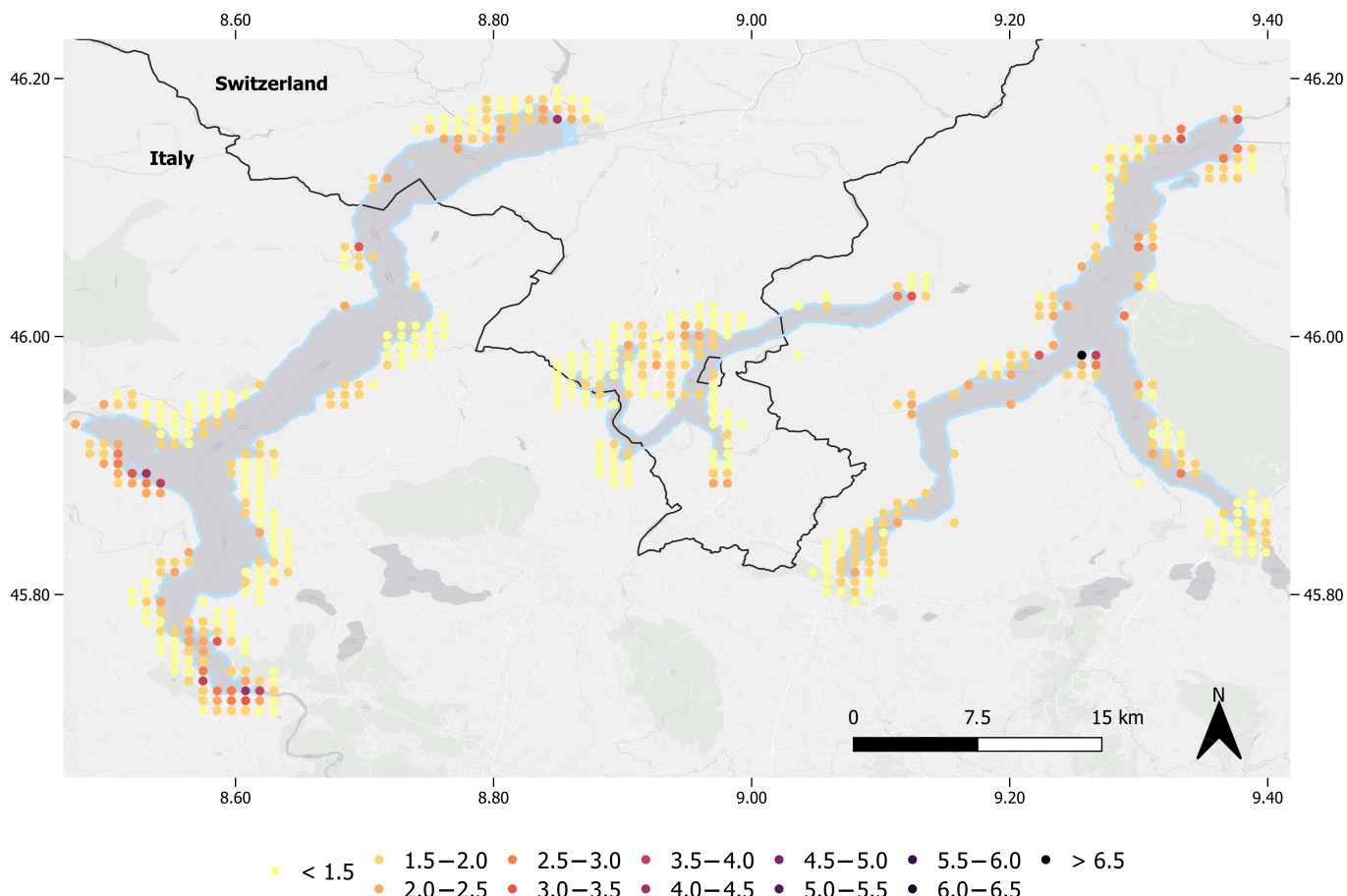

**Figure 6.** Map of the ratios between the 90th percentile of the time series of Facebook users' counts and the baseline population value. The ratio is computed at each location (Bing tile centre) in the dataset. Missing locations from the Bing tiles grid correspond to tiles with users' counts lower than 10 that were not available in the Facebook Population Maps dataset.

Figure 7 depicts the month of occurrence, with the distinction between weekends or weekdays, of the maximum values of the Facebook users' counts at each location. The maximum users' counts for the tourism hotspots of Maggiore and Como lakes, which showed the highest values for the previous indicator, were registered during weekends in the high tourism season. For the main urban centres, including Como and Lugano, the highest values generally occurred during the low tourism season (fall or winter) as well as weekdays, especially in the Lake Lugano areas.

The patterns highlighted by the indicator are consistent with the ones previously explored. The analysis of mean Facebook users' counts revealed global differences between lake areas that can be reasonably linked with their actual tourism attractiveness. Nevertheless, the combination of the information carried by the two proposed indicators provided additional insights into the interpretation of population patterns and anthropic pressure across the study region. To that end, the mapping of the indicators allowed us to point out the locations most suffering from occasional increases in human presence, providing a comparable measure for these pressures. Moreover, the distinction between tourism destinations and urban centres patterns was augmented by the display of both time and space dimensions of peak values for the Facebook users' counts.

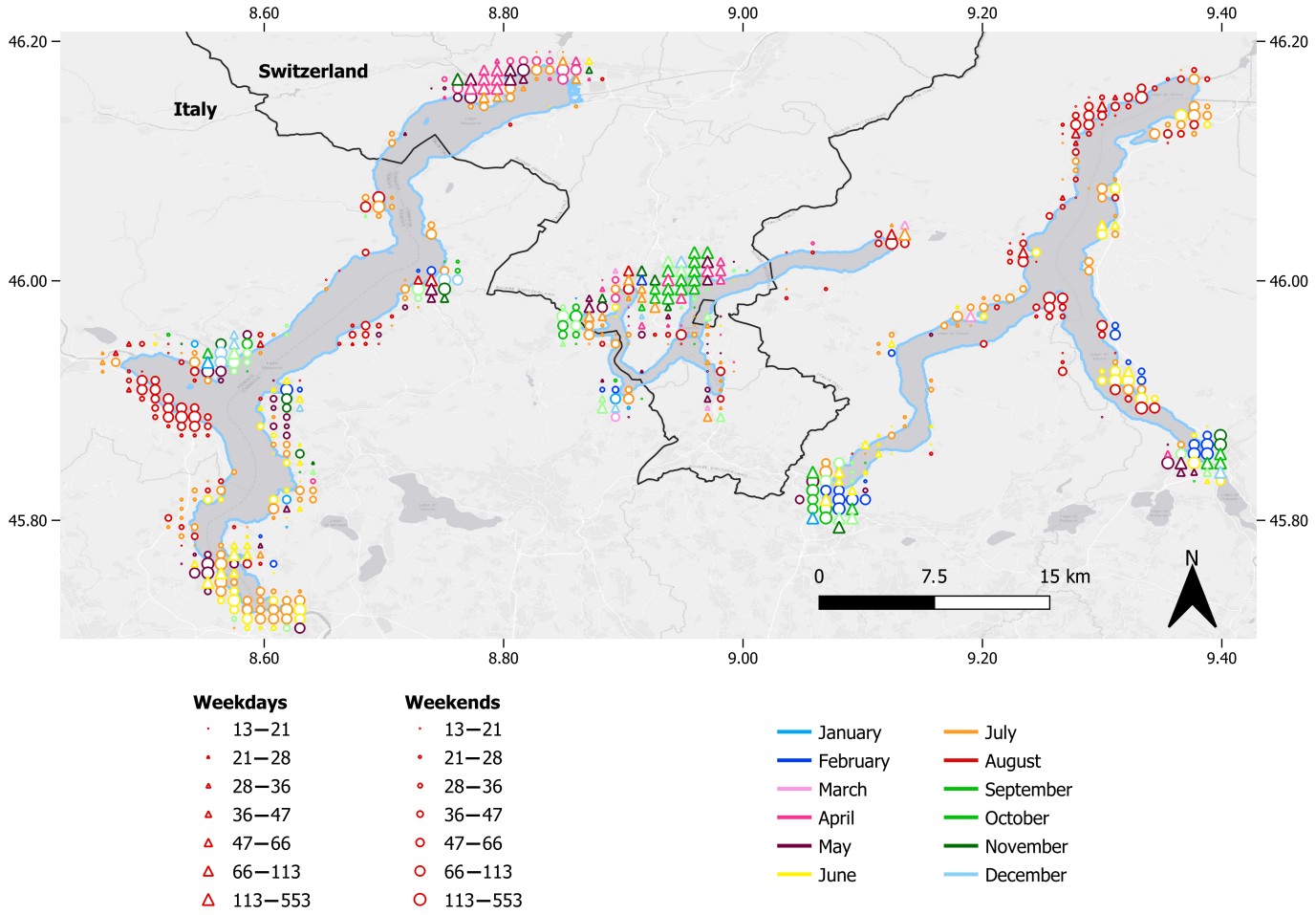

**Figure 7.** Map of the month of occurrence (marker colour) with distinction between weekdays (triangular marker) and weekends (circular marker) of the maximum values of the Facebook users' counts registered at each location (Bing tile centre). The marker size is proportional to the value. Missing locations from the Bing tiles grid correspond to tiles with users' counts lower than 10 that were not available in the Facebook Population Maps dataset.

## 5. Conclusions and Future Developments

In this paper, an assessment of population space–time patterns and anthropic pressure on lakes of the Insubria region between Northern Italy and Southern Switzerland was presented. Specifically, user-generated geodata retrieved from the Facebook Population Maps dataset was leveraged to investigate trends and differences in users' counts registered along lakes Maggiore, Como and Lugano in the period May 2020–August 2021. Numerical outcomes of exploratory analyses were included in graphs that allowed us to describe local patterns of users' presence and link them to tourism activities on lakeshores. Furthermore, the anthropic pressure, (intended here as systematic and significant increases of users' presence) affecting each location in the study region were assessed and located both in time and space through the design and computation of local indicators from the data.

Results outlined patterns similarity between lakes Maggiore and Como areas with significant increases in users' presence during weekends and in the high tourism season (average increase of 14.0% and 10.8%, respectively, in high season weekends than weekdays in the low season), thus suggesting patterns are strongly affected by leisure and tourism activities on their lakeshores. The indicator of anthropic pressure further confirmed the above disclosures by pointing out popular tourism locations as the most affected by increases in users' presence with respect to their baseline population (with peaks higher than 500% in some cases). Main urban centres of the two lakes resulted instead as less affected

by relative increases in users' presence with peaks mostly outside the high tourism season. Lake Lugano showed dissimilarity with respect to the other lakes. Indeed, the analysis of users' presence along its shores suggested patterns were affected mostly by the resident population as well as frontier workers that reach the area only during business days and hours (average increase of 6.6% compared to weekends during daytime). The indicator of anthropic pressure did not point out relevant users' presence hotspots during the high tourism season.

The user-generated geodata offered by the Facebook Population Maps provided extremely valuable space–time resolved information on human presence which—currently—would be impossible to retrieve from traditional population data sources such as national statistics bureaus. Concerns on the use of this dataset remain connected to both its low representativeness if compared with the actual resident population, and the impossibility of knowing how the different types of users (e.g., visitors, dwellers, transport and lake cleaning operators, etc.) influenced the counts. These groups might also have different impacts on lake pollution, which, however, cannot be directly considered using the Facebook Population Maps data. The above limitation might be partially overcome by integrating the analysis with traditional survey methods (e.g., interviews or questionnaires) to infer the use of the Facebook application on mobile devices among different user groups. However, the geographical extension and both the high population densities and tourism fluxes affecting the study area would have made such surveys hard to implement. Furthermore, the use of data different from the Facebook Population Maps is out of the scope of the presented research which aimed at investigating benefits and limitations of crowdsourced data for population patterns analysis. Finally, another marginal issue may be found in the roaming costs which may prevent mobile devices of users travelling across the national border to be equally recorded between Italy and Switzerland.

It is worth noticing that the COVID-19 pandemic may also have slightly affected population patterns, due to mobility restrictions that were in place both in Italy and Switzerland during the study period. In view of the above, future developments of the work will include additional tests on extended study periods as well as different study regions to outline the influence of these restrictions. Furthermore, the results of this study will be employed as a background for the implementation of numerical models to predict space–time population dynamics and the distribution of anthropic pressures. Finally, correlation analysis between human presence and lake-water quality parameters, including suspended solids and chlorophyll concentrations, will be carried out with the goal of estimating the influence of tourism on lakes water pollution. The above parameters are monitored by the SIMILE project through satellite observations and are already available as multi-temporal maps within the project outputs.

**Author Contributions:** Conceptualisation, Daniele Oxoli and Maria Antonia Brovelli; data curation, Alberto Vavassori and Daniele Oxoli; methodology, Alberto Vavassori, Daniele Oxoli and Maria Antonia Brovelli; project administration, Maria Antonia Brovelli; software, Alberto Vavassori and Daniele Oxoli; supervision, Maria Antonia Brovelli; validation, Alberto Vavassori, Daniele Oxoli; writing—original draft, Alberto Vavassori and Daniele Oxoli; writing—reviewed manuscript, Alberto Vavassori and Daniele Oxoli. All authors have read and agreed to the published version of the manuscript.

**Funding:** This research has been funded by the project "SIMILE" within the Interreg Co-operation Programme 2014–2020 (ID 523544).

**Institutional Review Board Statement:** Not applicable.

**Informed Consent Statement:** Not applicable.

**Data Availability Statement:** The paper is based on the datasets made available by Facebook through the Facebook Data for Goodinitiative (https://dataforgood.fb.com; accessed on 4 November 2021), of which access was kindly granted to the authors' institution for research purposes.

**Conflicts of Interest:** The authors declare no conflict of interest.

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
