# Peer review of "Population Space–Time Patterns Analysis and Anthropic Pressure Assessment of the Insubric Lakes Using User-Generated Geodata"

_ijgi, doi:10.3390/ijgi11030206_

Round 1
Reviewer 1 Report
I believe this paper is good and I accepted in the current form.
I have attached here the paper with some comments and questions.

Author Response
Response to Reviewer 1 Comments
Dear Reviewer #1,
thank you for appreciating our work. Your comments and questions were addressed in the below replies as well as in the manuscript (with purple text).
Point 1: (Abstract) “Please can you add more results numbers or percentages for this space-time pattern analysis.”
Response 1: We added summary percentages in the abstract at lines 11-12 and 15
Point 2: (Section 1) “What do you think about the fake information that you can get from social media? This will impact this analysis results? Did you calculate the error issued from that?”
Response 2: The data we used in the analysis is simply derived from the users’ device GPS collected by the Facebook app. Besides positional accuracies of any single user location log, which cannot be known due to privacy constraints, we believe the possibility of fake user positions is very limited and so its impact on the analysis. We added some details on individual user tracking in Section 2.1 which represents an intrinsic limitation of the considered data.
Point 3: (Section 1) “is nice to mention the name of authors and al; Name et al.,”
Response 3: Done at line 88
Point 4: (Section 1)“I am not sure that I am right about this following idea: I think that it could have bad impact in social media by making publicity for this nice area.”
Response 4: To that end, we tried to keep results disclosure neutral and we spell out clearly limitations of the proposed procedure. We believe that the analysis does not impact on reputation and visibility of these areas on social media, considering also the technical audience to which this paper is addressed. By doing so, we expect that any erroneous interpretation or use of the results would be avoided.
Point 5: (Figure 1) “Please change the km to the meter for the scale.”
Response 5: Due to the extension of the area, we believe that the km unit is more adequate to provide synoptic and readable insight into the dimension of the study region. We hope this is fine also with you.
Point 6: (Section 5) “Please support the conclusion with some numbers and percentages.”
Response 6: We added summary percentages for the three lakes in the Conclusions at lines 342-343, 346-347 and 352
Reviewer 2 Report
- The assumption of using two buffer zones to distinguish between tourism and residential activities needs to be further elaborated.
- The research design of using Facebook user count could be improved to consider individual users' activity patterns.
Author Response
Response to Reviewer 2 Comments
Dear Reviewer #2,
thank you for your valuable comments. We addressed them in the below replies as well as in the manuscript (with red text).
Point 1: “The assumption of using two buffer zones to distinguish between tourism and residential activities needs to be further elaborated”
Response 1:The assumption was better elaborated in Section 2.2 at lines 138-143
Point 2: “The research design of using Facebook user count could be improved to consider individual users' activity patterns”.
Response 2: We added an explanation on why this is not possible in Section 2.1 at lines 120-122
Reviewer 3 Report
The article is good and useful, but I see a few weak points in it.
If indeed, as the authors state, “Human activities are one of the main causes of lakes water pollution and eutrophication” (Line 1), then the indicators of pollution and eutrophication should be statistically significantly positively correlated with anthropogenic pressure. The authors are going to test this in further analyses. Those the analysis has not yet been done, the results are not known, but the article begins with the assumption that they will be positive. In this case, the assumption is expressed in the form of a peremptory statement - as a matter of course a priori. The authors refer to publications where it is written that this happens. However, is this the case for the Insubric and Southern Switzerland lakes? A separate question: have there any reliable primary data needed for such a check?
The magnitude of anthropic pressure in the areas surrounding the lakes was measured from users’ location records provided by the Facebook Data. However, it is not known whether the number of Facebook users reliably reflects the number of people with gadgets near the lake, and even more so the total number of any people near the lake. (The estimates given in section 2.3 are not very convincing). How many people go to the lake to swim without a phone, tablet or laptop? How many users prefer other social networks to Facebook? And how many do not use them at all? (To do this, you need to conduct statistical surveys in nature - to interview many people: do you have a gadget with you, is there a Facebook application in it). In addition, Facebook is unlikely to help separately determine the number of people involved in: 1) pollution, 2) cleaning the lake, and 3) not affecting these processes. Although only the first put pressure on the ecosystem.
It would be nice if the authors themselves wrote all this in the article in order to avoid an avalanche of scathing criticism from readers.
Author Response
Response to Reviewer 3 Comments
Dear Reviewer #3,
thank you for your insightful comments. We have addressed them one by one in the manuscript (with blue text) as well as in the below replies.
Point 1: “Human activities are one of the main causes of lakes water pollution and eutrophication” (Line 1), then the indicators of pollution and eutrophication should be statistically significantly positively correlated with anthropogenic pressure... Is this the case for the Insubric and Southern Switzerland lakes?”
Response 1: We added specific references in which a positive correlation between human-related activities and Insubric lakes pollution is reported (lines 48-50).
Point 2: “... Have there any reliable primary data needed for such a check?”
Response 2: Data that is intended to be used for this check in the future development of the work is provided by the mentioned SIMILE project. A clarification was added in the Conclusions at lines 382-384.
Point 3: “It is not known whether the number of Facebook users reliably reflects the number of people with gadgets near the lake... How many people go to the lake to swim without a phone, tablet or laptop?” How many users prefer other social networks to Facebook? And how many do not use them at all? (To do this, you need to conduct statistical surveys in nature - to interview many people: do you have a gadget with you, is there a Facebook application in it). How many users prefer other social networks to Facebook? And how many do not use them at all?”
Response 3: We pointed out this limitation by adding a specific comment in the Conclusion at lines 361-370.
Point 4: “... Facebook is unlikely to help separately determine the number of people involved in 1) pollution, 2) cleaning the lake, and 3) not affecting these processes. Although only the first put pressure on the ecosystem”
Response 4: We pointed out this limitation by adding specific comments in both Section 2 (lines 120-122, red text) and Section 5 (Conclusions) at lines 361-370.
Round 2
Reviewer 2 Report
Thank you for the revisions.